# Coronary Microvascular Dysfunction in Takotsubo Syndrome Assessed by Angiography-Derived Index of Microcirculatory Resistance: A Pressure-Wire-Free Tool

**DOI:** 10.3390/jcm10194331

**Published:** 2021-09-23

**Authors:** Jordi Sans-Roselló, Estefanía Fernández-Peregrina, Albert Duran-Cambra, Jose Carreras-Mora, Alessandro Sionis, Jesús Álvarez-García, Hector M. Garcia-Garcia

**Affiliations:** 1Acute and Intensive Cardiovascular Care Unit, Department of Cardiology, Hospital de la Santa Creu i Sant Pau, Biomedical Research Institute IIB-Sant Pau, 08041 Barcelona, Spain; aduranc@santpau.cat (A.D.-C.); asionis@santpau.cat (A.S.); 2Department of Medicine, School of Medicine, Universidad Autonoma de Barcelona, 08003 Barcelona, Spain; jalvarezg82@gmail.com; 3Section of Interventional Cardiology, MedStar Washington Hospital Center, EB 521,110 Irving St NW, Washington, DC 20010, USA; efernandezperegrina@gmail.com; 4Interventional Cardiology Unit, Department of Cardiology, Hospital de la Santa Creu i Sant Pau, Biomedical Research Institute IIB-Sant Pau, 08041 Barcelona, Spain; 5Acute and Intensive Cardiovascular Care Unit, Cardiology Department, Hospital del Mar, 08003 Barcelona, Spain; jcarrerasmora@psmar.cat; 6Advanced Heart Failure Unit, Department of Cardiology, IRYCIS, Hospital Universitario Ramón y Cajal, M-607, km. 9, 100, 28034 Madrid, Spain; 7Centro de Investigación Biomédica en Red de Enfermedades Cardiovasculares (CIBERCV), 28029 Madrid, Spain

**Keywords:** coronary microvascular dysfunction, takotsubo syndrome, angiography-derived index of microcirculatory resistance, left ventricular ejection fraction, cardiac biomarkers

## Abstract

Background: Coronary microvascular dysfunction (CMD) has been proposed as a key mechanism in Takotsubo syndrome (TTS). The non-hyperaemic angiography-derived index of microcirculatory resistance (NH-IMRangio) has been validated as a pressure-wire-free tool for the assessment of coronary microvasculature. We aimed to study the presence of CMD in TTS patients and its association with levels of cardiac biomarkers and systolic dysfunction patterns. Methods: We recruited 181 consecutive patients admitted for TTS who underwent cardiac angiography at a tertiary center from January 2014 to January 2021. CMD was defined as an NH-IMRangio ≥ 25. Plasma levels of NT-proBNP, high-sensitive cardiac troponin T (hs-cTnT) and the left ventricular ejection fraction (LVEF) by echocardiography were measured at admission. Results: Mean age was 75.3 years, 83% were women and median LVEF was 45%. All patients presented CMD (NH-IMRangio ≥ 25) in at least one epicardial coronary artery. The left anterior descending artery (LAD) showed higher median NH-IMRangio values than left circumflex (LCx) and right coronary arteries (RCA) (44.6 vs. 31.3 vs. 36.1, respectively; *p* < 0.001). NH-IMRangio values differed among ventricular contractility patterns in the LAD and RCA (*p* = 0.0152 and 0.0189, respectively) with the highest values in the mid-ventricular + apical and mid-ventricular + basal patterns. NT-proBNP levels, but not high-sensitive cardiac troponin T (hs-cTnT), were correlated with both the degree and the extent of CMD in patients with TTS. Lower LVEF was also associated with higher NH-IMRangio values. Conclusions: CMD is highly prevalent in patients admitted for TTS and is associated with both a higher degree of systolic dysfunction and higher BNP levels, but not troponin.

## 1. Introduction

Takotsubo syndrome (TTS) [1] is an acute and transient ventricular dysfunction with symptoms and electrocardiographic abnormalities that mimics acute myocardial infarction in the absence of obstructive epicardial coronary artery disease. The classical echocardiographic pattern in TTS is characterized by apical akinesis/hypokinesis with hypercontractile basal segments [2,3]. Although its prognosis is usually benign with a rapid systolic function improvement, some series have reported considerable morbidity and mortality in TTS patients [4,5].

Several mechanisms have involved in TTS, such as 1. a sudden surge in catecholamines and activation of the sympathetic nervous system [6,7]; 2. an acute and transient coronary microvascular dysfunction (CMD) [8,9,10,11,12,13,14]. CMD can be evaluated with different diagnostic methods: the index of microcirculatory resistance (IMR) is a quantitative and reproducible, wire-based method for invasively assessing the coronary microvascular function independent of the epicardial arteries [15]. IMR has prognostic implications in stable patients, acute coronary syndromes [16,17,18] and TTS patients [19,20,21,22,23,24]. However, it is not broadly used due to its cost, invasiveness and the need for a hyperaemic agent. Alternatively, an angiography-derived index of microcirculatory resistance (IMRangio), which is a novel angiography based index derived from the application of computational flow dynamics to three-dimensional modeling of the coronary artery and contrast flow by thrombolysis in the myocardial infarction (TIMI) frame count, has been recently introduced [25]. IMRangio has been validated as a pressure-wire-free for the assessment of coronary microvasculature in acute coronary syndromes (ACS), as well as in non-hyperaemic conditions (NH-IMRangio) [26,27,28].

Against this background, we carried out a study of the state of the microvasculature system in patients with TTS by measuring NH-IMRangio. The main objective of our study was to investigate and compare the degree and extent of CMD measured by NH-IMRangio in TTS patients. Moreover, we also investigated its correlation with different biomarkers, such as NT-proBNP, highly sensitive-cardiac troponin T (hs-cTnT) and left ventricular ejection fraction (LVEF).

## 2. Materials and Methods

### 2.1. Study Population

This is a retrospective observational study that recruited all consecutive patients admitted for TTS from January 2014 to January 2021 in a tertiary center in Barcelona (Spain). All inclusion criteria had to be met: (a) ≥ 18 years of age; (b) provision of signed, informed consent, (c) diagnosis of TTS according to modified Mayo Clinic criteria [29]; and (d) a coronary angiography performed (CAG) within the first 24 h of the onset of symptoms. Exclusion criteria were: (a) any condition preventing the use of NH-IMRAngio (e.g., severe calcification or vessel tortuosity, past history of coronary artery bypass grafting) or (b) newly diagnosed coronary artery disease in the same territory of the regional wall motion abnormality.

The study was conducted in accordance with the standards set by the “Declaration of Helsinki”.

### 2.2. Study Variables

The patient’s demographics, cardiovascular risk factors and clinical history were collected from medical reports at admission and discharge. ECG abnormalities were noted in the first ECG available. Serum levels of NT-proBNP and hs-cTnT from the first blood sample obtained were measured by electrochemiluminescence immunoassays on a Cobas e601 platform (Roche Diagnostics, Basel, Switzerland). The measurement ranges for NT-proBNP and hs-TnT were 5–35.000 pg/mL and 3–10.000 ng/L, and precision (expressed as CV) was ≤3.5% and ≤4.0%, respectively, according to the manufacturer. Other biochemical and hematological parameters were measured by standard procedures in the first blood test and arterial blood gas samples; the estimated glomerular function rate was calculated by the Modification of Diet in the Renal Disease Study equation (MDRD) [30]. The left ventricle ejection fraction (LVEF) was assessed by echocardiography using the biplane Simpson method at admission. Thereafter, the study population was classified into six groups based on the location of wall motion abnormalities by echocardiography: “apical limited”, “mid-ventricular limited”, “basal limited”, “mid-ventricular and apical”, “mid-ventricular and basal” and “others” if it did not belong to any of the previous categories. Management during the hospital stay was at the discretion of the treating physician. Percentage of endotracheal intubation with mechanical ventilation (EI-MV), use of inotropes, intra-aortic balloon pump (IABP) and renal replacement therapy (RRT) were also registered.

### 2.3. 3D-QCA, QFR and NH-IMRangio Assessment

The 3D-QCA analysis and QFR computation were performed in the CoreLab of the MedStar Washington Hospital Center using the QAngio XA 3D software package (Medis Suite 3.2.48.8, Medis, Leiden, The Netherlands). QFR analysis was performed by certified readers. Briefly, two angiographic projections, >25° apart, with the least foreshortening and minimum overlap of the main vessel and side branches were selected. In each projection, an end-diastolic frame was selected with ECG guidance to be used for analysis. One anatomical landmark of each projection was identified as the reference points for matching location information. Subsequently, we considered proximal and distal sites of the vessel, and vessel contours were automatically detected and manually corrected if necessary. The software reconstructed a 3D anatomical vessel model without its side branches for the 3D-QCA analysis and the QFR computation. Then we considered the number of frames (Nframes) required for contrast dye to travel from the proximal reference to the distal one.

NH-IMRangio was calculated using the validated formula [26,27]:$$\mathrm{NH}-\mathrm{IMRangio}=\mathrm{Pa}\left(\mathrm{resting}\right)*\mathrm{QFR}\left(\mathrm{resting}\right)*\left(\frac{\mathrm{Nframes}\left(\mathrm{resting}\right)}{\mathrm{fps}}\right)$$
where Pa(resting) is mean aortic pressure at resting; Nframes is the number of frames for contrast dye to travel from the proximal reference to the distal one; and fps is the frame-acquisition rate, set at 15 frames/second. We performed these measurements on the available coronary arteries of each patient. CMD was defined as a NH-IMRangio value ≥ 25 [31].

### 2.4. Statistical Analysis 

Results are presented as the mean (standard deviation) for continuous variables with a normal distribution, medians (interquartile range) for continuous variables with non-Gaussian distribution, and with counts and percentages for categorical variables.

We divided our cohort based on the median NH-IMRangio (≥50.6) and based on the number of arteries with a NH-IMRangio ≥ 25. We compared both of them using the x^2^ test or Fisher exact test for categorical variables. For continuous variables, they were analyzed by t-test or ANOVA in the case of a normal distribution and by Mann-Whitney U-test or Kruskal-Wallis test in the case of a non-normal distribution. Correlations were analyzed using Pearson if they presented a normal distribution and using Spearman if they had a non-Gaussian distribution. To study the correlation between microcirculatory dysfunction and biomarkers (NT-proBNP, hs-troponin T) and LVEF, the highest NH-IMRangio value of each patient was selected.

## 3. Results

We registered 181 patients diagnosed with TTS (Figure 1).

Most of the patients were women with a median age of 75.3 (65.3–81.8) years. Only 5% had previous coronary artery disease, while about 35% had a previously diagnosed psychiatric disorder such as depression and/or anxiety. The most common clinical symptoms at presentation were chest pain with diaphoresis, nausea and dyspnea. More than half of the patients referred a previous trigger, such as a stressful situation (physical or emotional), and around 30% of secondary forms were diagnosed. ST segment alterations and prolongation of the QT interval were the most common ECG findings. About 10% of the patients were admitted in a situation of cardiogenic shock requiring inotropic and/or vasoactive treatment. There was a need for mechanical ventilatory support (invasive/non-invasive) for over 15% of the patients. The left ventricle (LV) and other baseline characteristics of the patients are detailed in Table 1.

### 3.1. Evaluation of Microcirculatory Status in Epicardial Coronary Arteries

NH-IMRangio analysis was performed in at least one epicardial coronary artery in 166 patients (Figure 1). All patients presented at least one artery with NH-IMRangio values ≥ 25 (91.6% of LAD, 80.8% of LCx and 84.4% of RCA). QFR, TIMI frame count and NH-IMRangio were significantly different between arteries (*p* < 0.001). Values of both QFR, TIMI frame count and NH-IMRangio were higher in the LAD than in the other epicardial arteries. Detailed results of microcirculatory measurements in each artery are presented in Figure 2.

### 3.2. Evaluation of CMD in the Different Patterns of Wall Motion Abnormalities

We analyzed the NH-IMRangio of each epicardial artery based on the pattern of wall motion abnormalities that they presented by echocardiogram on admission. NH-IMRangio values in the LAD and in the RCA showed significant differences depending on the ventricular pattern (*p*: 0.0152 and *p*: 0.0189, respectively). On the other hand, the NH-IMRangio values in the LCx did not show statistically significant differences between motility patterns (*p*: 0.1869). Results are presented in Appendix A. 

Both in the apical limited pattern and in the mid-ventricular + apical pattern, NH-IMRangio values were different between arteries (*p*: 0.0066 and *p*: 0.0015, respectively) with higher values in LAD compared to LCx and RCA. In the other contractility patterns, no significant differences were found. Mid-ventricular + apical pattern and mid-ventricular + basal pattern presented the highest NH-IMRangio values in each artery. Detailed results of NH-IMRangio in patterns of wall motion abnormalities are shown in Figure 3.

### 3.3. Correlation of NH-IMRangio with Biomarkers & LVEF

The highest NH-IMRangio values of TTS patients presented a moderate positive correlation (Rho: 0.4845; *p* < 0.001) with the NT-proBNP values obtained at admission, while no correlation was found between NH-IMRangio and the hs-cTnT values (*p*: 0.1124). On the other hand, LVEF at admission showed a weak negative correlation with NH-IMRangio levels (Rho: 0.2606; *p*: 0.001). The scatter plots of these correlations are shown in Figure 4.

### 3.4. Differences between Patients by NH-IMRangio Value

The percentage of the patterns of wall motion abnormalities differed between groups (*p*: 0.04). In the NH-IMRangio ≥ 50.6 group, there was a higher percentage of patients with a mid-ventricular + apical pattern and a lower percentage with an apical limited pattern. NH-IMRangio ≥ 50.6 group presented higher NT-proBNP values and a lower LVEF than patients with lower values of NH-IMRangio. Both the atrial fibrillation and physical stressor were also higher in the higher NH-IMRangio group. Detailed differences between both groups are shown in Table 2.

### 3.5. Differences Based on Extension of Microvascular Dysfunction

NH-IMRangio of the three main coronary arteries was assessed in 131 patients. Patients with a greater number of coronary arteries with CMD (NH-IMRangio ≥ 25) showed a higher release of NT-proBNP (*p*: 0.0075). No differences were found in other analytical or hemodynamic parameters. In addition, there was a non-significant trend towards a higher percentage of mid-ventricular + apical patterns (*p*: 0.064) and a lower percentage of limited apical patterns (*p*: 0.054) among patients with a greater number of arteries with CMD. Detailed results are shown in Table 3. The results followed the same trend when all patients were examined regardless of the availability of analysis of all their coronary arteries (Appendix A).

## 4. Discussion

To our knowledge, this is the first study to address the status of the coronary microvasculature in the three coronary vessels using NH-IMRangio in patients with TTS. Our study has shown the following main findings: (1) TTS patients present an alteration of the coronary microvasculature with a predominance on the LAD territory; (2) the degree of microvascular involvement in patients with TTS is related to the different patterns of wall motion abnormalities; (3) the NH-IMRangio values are correlated with NT-proBNP levels and LVEF, but not with hs-cTnT in TTS patients.

First, it is controversial whether all patients with TTS present an alteration of the coronary microcirculation. In this setting, the evaluation of microcirculation in patients with TTS, both with semi-quantitative [8,10,32] and quantitative methods such as IMR [20], has reported inconclusive results. The study of coronary microvasculature in patients with TTS using IMR has been limited to the LAD [19,21,23,24], probably due to the invasiveness and costs of the procedure. Our study has shown a CMD in all patients with TTS. Since the NH-IMRangio calculation can be performed non-invasively, we were able to assess the microcirculation status of the three main epicardial coronary arteries in most patients. Our results showed that TTS patients could present signs of CMD in any of their coronary arteries, supporting the concept of global left ventricular (LV) involvement in patients with TTS [33,34]. Interestingly, the NH-IMRangio values differed between arteries with higher values in LAD. TTS mainly affects the LV, and the LAD irrigates the greater proportion of it, which could explain the higher values found in this territory. 

Second, we found significant differences in NH-IMRangio values in the different patterns of wall motion abnormalities in patients with TTS. Interestingly, patients with ventricular patterns that affected more than one territory (mid-ventricular + apical and mid-ventricular + basal) presented higher NH-IMRangio values compared to those in which only one ventricular territory was affected (apical, midventricular, or basal). These findings suggest that patients with a higher degree of CMD would show larger abnormalities in LV wall motility. Furthermore, patients with a greater number of arteries with evidence of CMD showed a trend towards a higher percentage of mid-ventricular + apical pattern and towards a lower percentage of the limited apical pattern. Although no statistical significance was obtained, this could suggest an eventual association between the extension of the CMD with more extensive patterns of ventricular contractility impairment. Further studies will be needed to confirm these results.

Finally, our results showed a correlation between the degree of CMD and both the release of NT-proBNP and LVEF in TTS patients. It is known that the release of NT-proBNP in acute coronary syndromes is produced by the stress of the LV wall when its filling pressures are raised. However, in TTS patients, NT-proBNP release has been correlated with the plasma catecholamines concentration and with LVEF [35]. We found that patients with higher NH-IMRangio presented higher NT-proBNP values and with lower LVEF, while patients with more extensive CMD (greater number of arteries affected) had higher NT-proBNP values without differences in LVEF. The small number of patients who presented NH-IMRangio values ≥ 25 in only one coronary artery (<10%) could have influenced the results. The elevation of hs-cTnT in patients with TTS is usually low compared to the degree of ventricular dysfunction [36]. In our study, the levels of hs-cTnT were slightly elevated in the vast majority of patients and did not correlate with the CMD parameters.

### Limitations

The indirect evaluation of the coronary microcirculation by NH-IMRangio is the main limitation of our study. NH-IMRangio is a surrogate, and it does not directly assess coronary microcirculation. Although NH-IMRangio has been validated as a pressure-wire-free alternative to IMR for the evaluation of coronary microvasculature [25,26,27,28,37,38,39] in different coronary diseases, this method has not been specifically validated in TTS patients.

Secondly, there is no specific cut-off point to define CMD in patients with TTS. In STEMI patients, where the main cause of CMD is microvascular obstruction, the cut-off point has been defined as an IMR > 40 [40]. However, since in TTS, the CMD is believed to be due to catecholaminergic toxicity along with endothelial dysfunction (functional rather than structural microvascular dysfunction), we believe that the widely spread cut-off point of 25 [31,41] could be more suitable for this population. Nevertheless, the lack of an established cut-off point in this setting could affect our results.

Moreover, despite having a relatively large sample of TTS patients, the low percentage of some of the contractility patterns represents a limiting factor to keep into account when interpreting the results of the current study. Lastly, NH-IMRangio was not compared with other angiography-derived indices recently developed [25,37,38,39] to contrast our findings about angiography-derived CMD in TTS patients.

## 5. Conclusions

In conclusion, the current study shows an alteration of the coronary microvasculature in patients with TTS. NH-IMRangio values are related to patterns of wall motion abnormalities as well as the degree of ventricular dysfunction in TTS. NT-proBNP, but not hs-cTnT, are associated with the degree and also with the extent of CMD in these patients. Further studies will be needed to confirm our findings and to validate the use of NH-IMRangio to assess CMD in TTS patients.

## Figures and Tables

**Figure 1 jcm-10-04331-f001:**
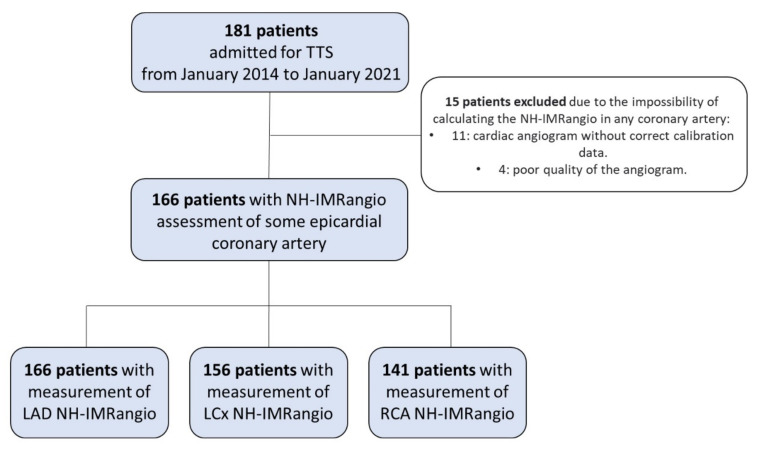
Patients flow chart. TTS: Takotsubo syndrome; NH-IMRangio: non-hyperaemic angiography-derived index of microcirculatory resistance; LAD: left anterior descending; LCx: left circumflex artery; RCA: right coronary artery.

**Figure 2 jcm-10-04331-f002:**
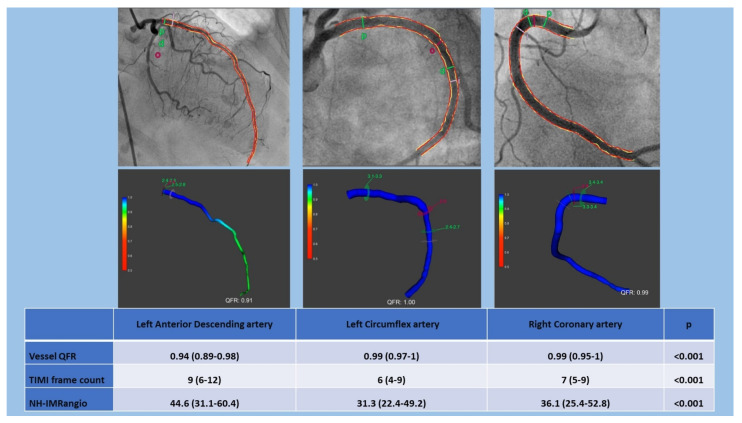
Hemodynamic measurements in patients with Takotsubo syndrome (TTS). QFR: quantitative flow ratio; NH-IMRangio: non-hyperaemic angiography-derived index of microcirculatory resistance.

**Figure 3 jcm-10-04331-f003:**
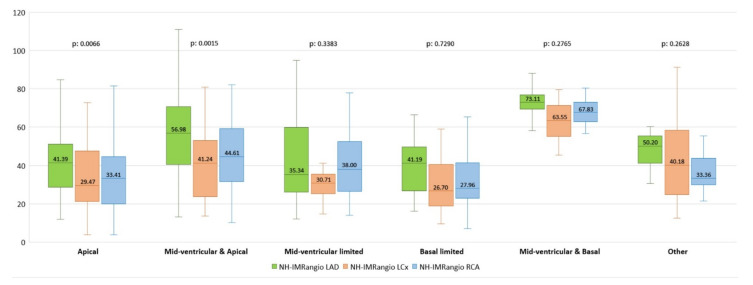
NH-IMRangio in the different wall abnormalities motility pattern in patients with Takotsubo syndrome (TTS). NH-IMRangio: non-hyperaemic angiography-derived index of microcirculatory resistance; LAD: left anterior descending; LCx: left circumflex artery; RCA: right coronary artery.

**Figure 4 jcm-10-04331-f004:**
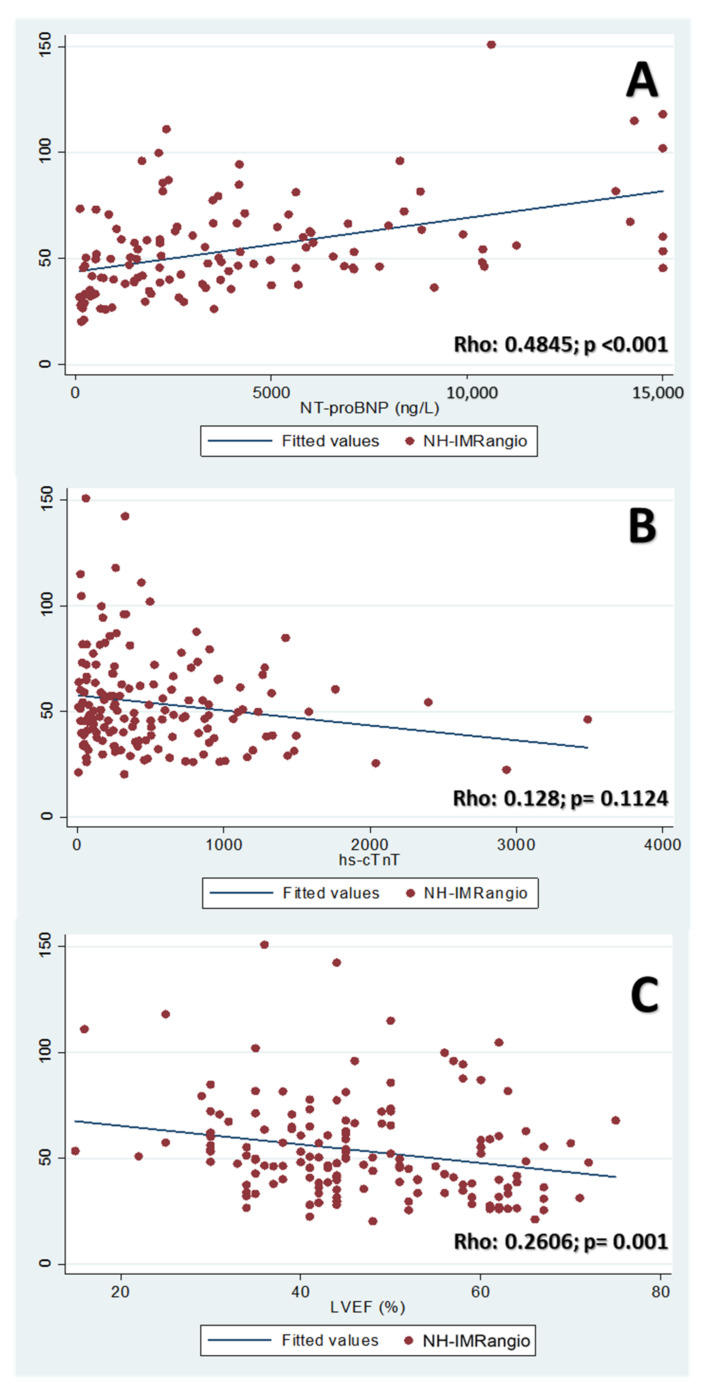
Correlations of NH-IMRangio with biomarkers and left ventricular ejection fraction (LVEF). Panel (**A**): Correlation between NT-proBNP values (pg/mL) and NH-IMRangio. Panel (**B**): Correlation between high-sensitive cardiac troponin T values (ng/L) and NH-IMRangio. Panel (**C**): Correlation between LVEF values (%) and NH-IMRangio.

**Table 1 jcm-10-04331-t001:** Clinical, ECG, biochemical, echocardiographic and therapeutic characteristics of the study population at admission.

	All Patients (*n* = 181)
Age (years)	75.3 (65.3–81.8)
Female gender	83.4
Hypertension	68.5
Diabetes mellitus	24.9
Dyslipidemia	46.9
Current smoker	17.1
Body mass index (kg/m^2^)	24.7 (22.2–27.5)
Previous coronary artery disease	5.0
Previous psychiatric disorder	34.3
Chronic kidney disease	10.5
TAKOTSUBO-RELATED DATA	
Clinical presentation:	
- Chest pain	69.3
- Vegetative symptoms	42.2
- Dyspnea	50.0
- Palpitations	3.6
- Syncope	6.6
- Cardiac arrest	6.0
Previous stressful situation	57.5
Secondary form of TTS	31.9
Systolic blood pressure (mmHg)	128 (114–146)
Heart rate (bpm)	85 (75–100)
Killip-Kimball class:	
- I	64.1
- II	14.9
- III	11.1
- IV	9.9
Left ventricular ejection fraction (%)	45 (35–55)
Patterns of wall motion abnormalities:	
- Apical limited	48.1
- Mid-ventricular and Apical	25.4
- Mid-ventricular	13.8
- Basal limited	6.1
- Mid-ventricular and Basal	2.8
- Other	3.8
ECG DATA	
Sinus rhythm	87.9
ST-segment elevation	54.7
ST-segment depression	46.4
Negative T waves	34.9
Long QT interval	45.3
QT interval (msec)	450 (430–500)
BLOOD TEST DATA	
pH	7.36 (7.27–7.43)
Lactate (mmol/L)	2.7 (1.2–5.0)
Hemoglobin (g/L)	129 (116–141)
hs-cTnT (ng/L)	250.5 (80.5–656.0)
eGFR (ml/min/1.73 m^2^)	76.0 (54.3–89.7)
NT-proBNP (pg/mL)	3300 (1318–6955)
MANAGEMENT	
Need for non-invasive ventilation	13.3
Need for invasive ventilation	13.8
Use of inotropes	13.3
Intra-aortic counterpulsation balloon	2.2
Renal replacement therapy	5.0

Continuous variables are expressed as median (IQR) and categorical data as %. Abbreviations: kg/m2: kilograms/square metres; TTS: Takotsubo syndrome; mmHg: millimetre of mercury; bpm: beats per minute; msec: milliseconds; mmol/L: millimoles per litre; g/L: grams per litre; hs-cTnT: high-sensitive Troponin T; ng/L: nanograms per litre; pg/mL: nanograms per litre; eGFR: estimated glomerular filtration rate; mL/min/1.73 m^2:^ millilitres per minute/1.73 square metres; pg/mL: nanograms per litre; IQR: interquartile range.

**Table 2 jcm-10-04331-t002:** Differences between patients based on NH-IMRangio values (*n*: 166 patients).

Characteristic	NH-IMRangio < 50.6 (*n*: 83)	NH-IMRangio ≥ 50.6 (*n*: 83)	*p*-Value
Age (years)	75.6 (65.0–82.4)	74.4 (64.9–80.6)	0.7578
Female gender	79.5	86.8	0.214
Prior physical stressful trigger	44.6	26.5	0.015
Prior emotional stressful trigger	13.3	27.7	0.021
Patterns of wall motion abnormalities:			0.04
- Apical limite	60.2	39.8	0.008
- Mid-ventricular + Apical	16.9	34.9	0.008
- Mid-ventricular limited	12.1	12.1	1
- Basal limited	7.2	4.8	0.514
- Mid-ventricular + Basal	0	2.4	0.155
- Other	3.6	6.0	0.469
SBP (mmHg)	134 (120–150)	124 (110–140)	0.056
Heart rate (bpm)	85 (73–100)	85 (77–96)	0.719
Killip class at admission:			0.499
- I	62.7	67.5	0.515
- II	16.9	14.5	0.669
- III	13.3	7.2	0.201
- IV	7.2	10.8	0.417
Atrial fibrillation	2.4	9.6	0.05
LVED-pressure (mmHg)	17 (13–23)	19 (12–25)	0.5957
LVEF (%)	47 (42–59)	42.5 (35–50)	0.0047
pH	7.36 (7.29–7.42)	7.37 (7.26–7.44)	0.6097
Lactate (mmol/L)	2.6 (1.2–3.1)	2.6 (0.9–3.4)	0.7478
Hemoglobin (g/L)	124 (111–140)	133.5 (122–143)	0.007
NT-proBNP (pg/mL)	2716.5 (935.5–5315.5)	4198 (2154–8390)	0.011
hs-cTnT (ng/L)	250 (79–630)	247 (77–661)	0.8185

Continuous variables are expressed as median (IQR) and categorical data as %. SBP: systolic blood pressure; mmHg: millimeter of mercury; bpm: beats per minute; LVED-pressure: end-diastolic left ventricular pressure; LVEF: left ventricle ejection fraction; mmol/L: millimoles per liter; g/L: grams per liter; pg/mL: nanograms per liter; hs-cTnT: high-sensitive Troponin T; ng/L: nanograms per liter; IQR: interquartile range.

**Table 3 jcm-10-04331-t003:** Differences in patients based on the number of arteries with CMD (*n*: 131 patients).

	One Artery Affected (*n*: 11)	Two Arteries Affected (*n*: 38)	Three Arteries Affected (*n*: 82)	*p*
Age (years)	72.9 (67.5–81.3)	70.5 (64.7–77.3)	75.6 (64.3–83.0)	0.1467
Female gender	72.7	79.0	86.6	0.363
Prior physical stressful trigger	36.3	39.5	36.6	0.952
Prior emotional stressful trigger	18.2	21.1	20.7	0.978
Patterns of wall motion abnormalities:				0.296
- Apical limited	72.7	55.3	42.7	0.054
- Mid-ventricular + Apical	15.8	18.2	35.4	0.064
- Mid-ventricular limited	0	13.2	12.2	0.455
- Basal limited	9.1	7.9	4.9	0.741
- Mid-ventricular + Basal	0	0	2.4	0.545
- Other	0	7.9	2.4	0.275
SBP (mmHg)	113 (95–133)	111 (94–143)	128 (115–143)	0.5594
Heart rate (bpm)	88.5 (77–120)	84.5 (74–95)	81 (73–93)	0.4548
Atrial fibrillation	9.1	2.6	7.3	0.554
LVED-pressure (mmHg)	16.5 (14–23.5)	16.5 (12–23)	18 (12.5–24.5)	0.9201
LVEF (%)	48 (41–58)	44 (35–58)	44 (38–52)	0.7174
pH	7.32 (7.30–7.36)	7.29 (7.26–7.37)	7.42 (7.34–7.44)	0.1820
Lactate (mmol/L)	2.4 (0.4–2.9)	2.9 (0.9–3.5)	2.3 (0.8–3.1)	0.4237
Hemoglobin (g/L)	121 (100–130)	133 (122–143)	130 (117–140)	0.0741
NT-proBNP (pg/mL)	200 (127–3907)	2799 (1414–5786)	3650 (1924–7400)	0.0075
hs-cTnT (ng/L)	215 (45–487)	216 (91–628)	260 (64–643)	0.8283

Continuous variables are expressed as median (IQR) and categorical data as %. CMD: coronary microcirculatory disease; SBP: systolic blood pressure; mmHg: millimeter of mercury; bpm: beats per minute; LVED-pressure: end-diastolic left ventricular pressure; LVEF: Left ventricle ejection fraction; mmol/L: millimoles per liter; g/L: grams per liter; pg/mL: nanograms per liter; hs-cTnT: high-sensitive Troponin T; ng/L: nanograms per liter; IQR: interquartile range.

## Data Availability

The data presented in this study are available on request from the corresponding author.

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
