# Peer review of "Coronary Microvascular Dysfunction in Takotsubo Syndrome Assessed by Angiography-Derived Index of Microcirculatory Resistance: A Pressure-Wire-Free Tool"

_jcm, 2021, doi:10.3390/jcm10194331_

Round 1

Reviewer 1 Report

Dear Authors, 

the paper is well written, focused and interesting. English is good.
Bibliography is up to date. Maybe conclusion are too short and the mechanism of coronary microvascular dysfunction in TTS are not particularly underlined.

Reviewer 2 Report

In the present paper Sans-Rosello et al. attempt to characterize coronary microvascular dysfunction in a retrospective cohort of 161 patients diagnosed with Takotsubo syndrome by using a nontraditional non-hyperaemic angiography-derived index of microcirculatory resistance (IMRangio). The authors used a surrogate for microvascular function (non-hyperemic IMRangio) by frame count analysis of baseline coronary angiographic images without hyperemic stimulation (eg without application of adenosine). The authors claim that all patients had coronary microvascular dysfunction (CMVD) based on non-hyperemic IMRangio with cut-off of >=25. It is important to note, that IMR of 25 is a well-accepted cut-off for the diagnosis of CMVD for conventional IMR done by catheter-based assessment in the presence of maximum hyperemia, it is certainly not a well-established value for IMRangio and especially not for non-hyperemic IMRangio. The authors noted higher non-hyperemic IMRangio values with more extensive ventricular involvement (in patterns with basal/midventricular and mid/apical ventricular pattern). Non-hyperemic IMRangio values correlated with NT-proBNP, and inversely correlated with echocardiographic LVEF. These are interesting findings; however, I am still concerned that the method used is not properly validated for the assessment of coronary microvascular function.

Specific comments:

1) How did the authors pick the NH-IMRangio cut-off of 25? Is there any scientific validation for this? The paper by Scarsini et al (REF 28) suggests an algorithmic approach to define CMVD with NH-IMRangio if 1) the NH-IMRangio >90 or 2) with further validation by vasodilator assessment if NH-IMRangio is between 30 and 90. Was there any attempt to do adenosine IMRangio in these patients? Even this algorithm is limited to one publication only, it would require further validation. In this report the correlation between conventional IMR and NH-IMRangio was moderate at least (rho = 0.64). Can the authors provide more data for validation?

2) The current title does not make sense, consider replacing “assess by” with “assessed by”

3) There abstract has IMR-angio values, are these mean or median, please label them correctly or add standard deviations

4) Page 2 line 65, REF 25 is incorrect, this paper does not mention IMRangio at all.

5) Per the Methods section, continuous variables were analyzed by t-test or Mann-Whitney U-test. However, many analyses contained multiple groups, please clarify the statistical method used to analyze these.

Author Response

请参阅附件

Round 2

Reviewer 2 Report

Unfortunately, the authors continue to fail to provide sufficient rationale for the arbitrary cut-off selection of 25 for normal IMRangio values. While the provided reference by Mejia-Renteria suggests that there is linear correlation between IMRangio and conventional IMR, this correlation is far from perfect (r2 of 0.49) and key figure from the mentioned paper suggests that IMR and IMRangio values are not interchangeable (eg a conversion factor might be needed to calculate one from the other). Based on this I don’t think the conclusion points are justified.